Construction and validation of an RNA-binding protein-associated prognostic model for colorectal cancer

http://orcid.org/0000-0002-1429-8915 Miao Yandong 1 miaoyd19@lzu.edu.cn
Zhang Hongling 2
Su Bin 3
Wang Jiangtao 1
Quan Wuxia 4
Li Qiutian 3
Mi Denghai 1 5 mi.dh@outlook.com
1 The First Clinical Medical College, Lanzhou University , Lanzhou, Gansu , China
2 Cancer Ward, Palliative Medical Center, New Kunhua Hospital , Kunming, Yunnan , China
3 Department of Oncology, The 920th Hospital of the Chinese People’s Liberation Army Joint Logistic Support Force , Kunming, Yunnan , China
4 Qingyang People’s Hospital , Qingyang, Gansu , China
5 Gansu Academy of Traditional Chinese Medicine , Lanzhou , China
Albertini Maria Cristina
Electronic publication date: 2021 Apr 5
Publication date: 2021
Volume: 9
Electronic Location ID: e11219
Received 2020 Dec 30; Accepted 2021 Mar 15
Copyright: © 2021 Miao et al.
Copyright year: 2021
Copyright holder: Miao et al.
License: This is an open access article distributed under the terms of the Creative Commons Attribution License, which permits unrestricted use, distribution, reproduction and adaptation in any medium and for any purpose provided that it is properly attributed. For attribution, the original author(s), title, publication source (PeerJ) and either DOI or URL of the article must be cited.
License URL: https://creativecommons.org/licenses/by/4.0/

Keywords: Colorectal cancer, RNA-binding proteins, Prognostic model, Bioinformatic analysis, TCGA, GEO

Funding: The authors received no funding for this work.

==============================
Colorectal cancer (CRC) is one of the most prevalent and fatal malignancies, and novel biomarkers for the diagnosis and prognosis of CRC must be identified. RNA-binding proteins (RBPs) are essential modulators of transcription and translation. They are frequently dysregulated in various cancers and are related to tumorigenesis and development. The mechanisms by which RBPs regulate CRC progression are poorly understood and no clinical prognostic model using RBPs has been reported in CRC. We sought to identify the hub prognosis-related RBPs and to construct a prognostic model for clinical use. mRNA sequencing and clinical data for CRC were obtained from The Cancer Genome Atlas database (TCGA). Gene expression profiles were analyzed to identify differentially expressed RBPs using R and Perl software. Hub RBPs were filtered out using univariate Cox and multivariate Cox regression analysis. We used functional enrichment analysis, including Gene Ontology and Gene Set Enrichment Analysis, to perform the function and mechanisms of the identified RBPs. The nomogram predicted overall survival (OS). Calibration curves were used to evaluate the consistency between the predicted and actual survival rate, the consistency index (c-index) was calculated, and the prognostic effect of the model was evaluated. Finally, we identified 178 differently expressed RBPs, including 121 up-regulated and 57 down-regulated proteins. Our prognostic model was based on nine RBPs (PNLDC1, RRS1, HEXIM1, PPARGC1A, PPARGC1B, BRCA1, CELF4, AEN and NOVA1). Survival analysis showed that patients in the high-risk subgroup had a worse OS than those in the low-risk subgroup. The area under the curve value of the receiver operating characteristic curve of the prognostic model is 0.712 in the TCGA cohort and 0.638 in the GEO cohort. These results show that the model has a moderate diagnostic ability. The c-index of the nomogram is 0.77 in the TCGA cohort and 0.73 in the GEO cohort. We showed that the risk score is an independent prognostic biomarker and that some RBPs may be potential biomarkers for the diagnosis and prognosis of CRC.

Introduction

Colorectal cancer (CRC) is the fourth most lethal cancer, with about 900,000 deaths annually. It is the second most common cancer in women and the third most common cancer in men (Dekker et al., 2019). CRC has a complicated biological progression affected by multiple factors, such as lifestyle, obesity and environmental factors. Its development involves profound changes at various molecular levels, including in the transcriptome, genome, epigenome and proteome (Murphy, Jenab & Gunter, 2018). Despite significant advances in diagnostic and therapeutic approaches, the average 5-year survival rate of CRC is 48% (Sharma, 2020); the 5-year survival rate for patients with metastatic CRC is only 10% (Lafitte, Sirvent & Roche, 2020). CRC diagnosis typically relies on imaging evaluation, molecular cancer biomarkers and histopathological examination. It is difficult to obtain an early diagnosis for CRC (Jung et al., 2020; Luo et al., 2019), leading to a high mortality rate for CRC patients. Consequently, we need to understand the molecular mechanisms of CRC better and explore more efficient early screening and diagnosis methods to improve the treatments and quality of life for CRC patients.

RNA-binding proteins (RBPs) are generally considered to be proteins that bind RNA through one or more globular RNA-binding domains and alter the fate or function of the bound RNA (Hentze et al., 2018). RBPs are crucial to RNA metabolism and regulate the spatial, temporal and functional dynamics of RNA. Altered RBP expression affects cell physiology and the RNA processes from pre-mRNA splicing to protein translation. Current genetic and proteomic data and results from animal models suggest that RBPs are precipitated in many human diseases, from neurological disorders to cancer (Lukong et al., 2008). Approximately 1,542 RBP genes have been identified through whole-genome screening of the human genome (Gerstberger, Hafner & Tuschl, 2014). These RBPs contribute to genetic regulation over a wide range of cellular and developmental processes (Kong & Lasko, 2012). Previous research has demonstrated that RBPs, including the human pumilio proteins, PUM1 and PUM2, affect human disease genes related to cancer and neurological and cardiovascular diseases (Bohn et al., 2018).

Previous studies have shown that RBPs are critical modulators of transcription and translation, which are often dysregulated in cancer and are associated with cancer transformation, tumorigenesis and poor prognosis (Dang et al., 2017; Kim et al., 2015; Wang et al., 2019b). Recent research has shown that RBPs regulate the dynamic balance of proliferation, migration, differentiation and senescence of intestinal epithelial cells, interacting with crucial pathways in CRC (Chatterji & Rustgi, 2018). A limited number of RBPs have been well-studied and have been found to play vital roles in regulating the initiation and progression of cancer (Van Kouwenhove, Kedde & Agami, 2011). For example, CRD-BP is necessary to induce both beta TCP1 and c-Myc through beta-catenin signaling in CRC cells (Noubissi et al., 2006). Heterogeneous ribonucleoprotein A0 promotes excessive mitosis and the growth of CRC cells (Konishi et al., 2020). Two RBPs have been explored that had obvious effects on the metastasis and overall survival (OS) of CRC patients (Zhou & Guo, 2018). However, RBPs regulate CRC progression through poorly understood mechanisms. Previous studies have shown that twelve identified RBPs may be promising predictors of CRC but no further verification at the protein expression level (Fan et al., 2020). Zhang et al. (2020) was constructed a prognostic risk score model based on four RBPs and explore their prognosis value for CRC patients. But no further analysis of the potential prognosis proteins co-expressed with RBPs (Zhang et al., 2020). A comprehensive functional study of RBPs is needed to better understand their role in CRC. In the current study, mRNA sequencing and clinical data of CRC were obtained from The Cancer Genome Atlas (TCGA) database. We illuminated abnormally expressed RBPs from normal and cancer samples through a series of bioinformatics analyses and comprehensively identified their potential functions and molecular mechanisms. Besides, we also identified several potential prognosis proteins co-expressed with RBPs. Our study identified some CRC-related RBPs that assist in understanding the molecular mechanisms of CRC progression. We developed a novel risk profile as an independent prognostic biomarker for risk stratification in CRC patients. Our research design and analysis flow chart are shown in Fig. 1.

Figure 1 A flow chart of the study design and analysis.

Materials and Methods

Data sources

mRNA sequencing and clinical data for CRC were downloaded from the TCGA database (Data Release 24.0-7 May 2020, https://portal.gdc.cancer.gov/). Detailed data selection criteria described in our previous study: filter criteria of mRNA sequencing: (1) the keywords of cases are (Primary Site) “colon, rectosigmoid junction and rectum”, (Program) “TCGA”, (Project) “TCGA-COAD, TCGA-READ”, (Disease Type) “Adenomas and Adenocarcinomas”. (2) The keywords of files are (Data Category) “Transcriptome Profiling”, (Data Category) “Gene Expression Quantification”, (Experimental Strategy) “RNA-Seq”, (Workflow Type) “HTSeq-FPKM”, filter criteria of clinical data: (Data Category) “Clinical”, (Data Format) “BCR XML”) (Miao et al., 2020). We obtained 692 mRNA expression profiles of CRC. Among them, 51 (7.4%) samples were normal and 641 samples (92.6%) were cancerous. The TCGA-CRC cohort consisted of 624 patients. The clinicopathological features of the patients are listed in Table S1. The GEO dataset was chosen mainly as a validation dataset to prove the accuracy of analysis results based on TCGA dataset. The data access (GSE39582, GSE87211) was obtained from the Gene Expression Omnibus (GEO, https://www.ncbi.nlm.nih.gov/geo/) for verification studies. The GSE39582 dataset consisted of 585 samples, including 19 colon mucosa samples and 566 stage I–IV colon adenocarcinoma samples. The GSE87211 dataset included 363 samples, including 160 rectal mucosa samples and 203 rectal tumor samples (Hu et al., 2018; Marisa et al., 2013). We merged the two datasets through a Perl script. Raw data from gene chips were normalized using the RMA algorithm provided by the R-package “limma” (Ritchie et al., 2015). Perl script and the R-package “sva” were used to merge microarray data and decrease heterogeneity between the two studies (Zhang et al., 2019).

Identification of differential expression RBPs

RNA binding proteins (RBPs) were obtained from the literature (Gerstberger, Hafner & Tuschl, 2014). Perl software was used to extract and integrate the data. We used the Wilcox Test in R-package “limma” to screen differentially expressed RBPs. False-positive discovery (FDR) < 0.05 and Log2 | (fold change, FC) | > 1 was set as the cutoffs. R-package “pheatmap” was used to analyze Bidirectional hierarchical clustering and draw a heat map.

Functional enrichment analysis of RBPs

Functional annotation and enrichment analysis of up-regulated and down-regulated RBPs were based on the R-packages “clusterProfiler”, “org.Hs.eg.db”, “ggplot2” and “enrichplot”, which classified Gene Ontology (GO) to the Biological Processes (BP), Cellular Components (CC) and Molecular Functions (MF). FDR < 0.05 was set as the cutoff. Kyoto Encyclopedia of Genes and Genomes (KEGG) pathways analysis was performed by Gene Set Enrichment Analysis (GSEA). It was considered statistically significant that the number of permutations was set to 1,000, P-value < 0.05 and FDR < 0.25.

Protein–protein interaction network constructed and module selection

The STRING database (http://www.string-db.org/) (Von Mering et al., 2003) was used to evaluate the protein–protein interaction (PPI) information from differently expressed RBPs. Cytoscape software (version 3.7.2) was used to construct and visualize the PPI network. The essential genes and modules were selected in the PPI network using the Molecular Complex Detection (MCODE) plug-in with MCODE and node count numbers more significant than five (Bader & Hogue, 2003).

Construction prognosis model of RBPs and survival analysis

Univariate Cox regression analysis was used to screen out the RBPs that had a significant relationship with the OS of CRC patients (P < 0.05). We used multivariate Cox regression analysis to construct an optimal model of RBPs. We used Kaplan–Meier analysis and the log-rank test (P < 0.05) to assess survival. Each patient’s risk score was calculated using the following formula: Risk score = ∑n=1n⁡Coefficient(n)×geneexpression(n), Coefficient(n) representing the regression coefficient and geneexpression(n) indicating the relative expression levels of each prognosis-associated RBPs normalized by z-score in the prognostic risk score model, respectively (Liu et al., 2019; Zhang et al., 2020). Survival information was obtained from the clinical data of the aforementioned samples. The median risk score was chosen as the cutoff value for the CRC cohort dichotomy, and the CRC patients were categorized into high and low-risk groups. We used the R-packages “survminer” and “survival” to draw the survival curve according to the high and low-risk values. The Receiver Operating Characteristic (ROC) curve was used to examine the specificity and sensitivity of survival prediction by the protein characteristic risk score and was drawn using the R-package “survival ROC”. The area under curve (AUC) value is an indicator of prognostic accuracy (Sachs, 2017). We drew a heat map, risk curve and survival state diagram based on the different risk scores of our patients. Independent prognostic proteins were identified using univariate and multivariate Cox hazard regression analysis. A total of 758 CRC patient samples with prognostic information from the GSE39582 and GSE87211 datasets were used as verification cohorts to verify the prognostic model’s predictive ability. Protein co-expression was analyzed using the R-packages “ggalluvial”, “ggplot2” and “dplyr”.

Immunohistochemistry analysis

We used the Human Protein Atlas (HPA, http://www.proteinatlas.org/) database to explore the expression level of the hub RBPs in the normal and tumor samples. Analysis of each protein and its corresponding cancer type in patients, using Immunohistochemistry (IHC) analysis based on tissue microarrays, is presented for a majority of the protein-coding genes. IHC staining was conducted according to prior published literature: the staining index scores were distributed as follows: staining intensity (negative: 0; Weak: 1; Moderate: 2; Strong: 3), positive staining (0: <5%; 1: 5–25%; 2: 26–50%; 3: 51–75%; 4: >75%). The staining index score was computed through multiplying the staining intensity score by the positive staining score, which ranged from 0 to 12 (Zhang et al., 2020).

Construction the nomogram

We used the expression level of RBPs by the R-package “rms” to construct the nomogram and predict the likelihood of OS. Calibration curves were used to estimate the consistency between predicted and actual survival, and the performance of the model in predicting prognosis was evaluated by the consistency index (C-index). C-index values of 0.5 and 1.0 represent a random probability and an excellent ability to predict survival, respectively.

Results

Identification of differently expressed RBPs in CRC patients

One thousand, five hundred and forty-two RBPs were used in our study. One hundred and seventy-eight differently expressed RBPs were excluded, and 121 up-regulated and 57 down-regulated RBPs were included in our study (Fig. 2).

Figure 2 Differentially expressed RBPs in Colorectal cancer.

(A) Heatmap: higher expression RBPs are shown in red; lower expressions RBPs are shown in green. (B) Volcano diagram: Each point represents a protein: down-regulated (blue), up-regulated (red), and no significant (black). RBPs, RNA-binding proteins.

Identification of involved signaling pathways of the differently expressed RBPs

We divided the selected RBPs into up-or down-regulated expression groups to explore their function and mechanisms. Functional enrichment analysis was conducted using R software. The results showed that 105 GO terms of BP, 21 GO terms of CC and 48 GO terms of MF were significant in the up-regulated group (FDR < 0.05). Seventy-three GO terms of BP, three GO terms of CC and 25 GO terms of MF were significant in the down-regulated group (FDR < 0.05). The up-regulated RBPs were mainly enriched in ncRNA metabolic process during BP analysis, ncRNA processing, nucleic acid phosphodiester bond hydrolysis, RNA phosphodiester bond hydrolysis and ribonucleoprotein complex biogenesis (Fig. 3A). The down-regulated differently expressed RBPs were significantly enriched in mRNA processing, RNA splicing, defense response to a virus, response to a virus and translation regulation (Fig. 3B). During the CC analysis, the up-regulated RBPs were enriched in the nucleolar part, cytoplasmic ribonucleoprotein granule, ribonucleoprotein granule, nuclear envelope and chromatin (Fig. 3A). In contrast, the down-regulated RBPs were enriched in the endolysosome membrane, apical dendrite and endolysosome (Fig. 3B). In terms of MF, the up-regulated differently expressed RBPs were notably enriched during catalytic activity, when acting on RNA, and during nuclease activity, ribonuclease activity, endonuclease activity and tRNA binding (Fig. 3A). The down-regulated differently expressed RBPs were significantly enriched in mRNA binding, AU-rich element-binding, mRNA 3′-UTR AU-rich region binding, mRNA 3′-UTR binding double-stranded RNA binding (Fig. 3B). The top-10 terms of BP, CC and MF are shown in Fig. 3. The detailed information of GO the terms is shown in Table S2. According to the GSEA analysis, KEGG pathways of differently expressed RBPs showed that 66/165 gene sets were up-regulated in phenotype h; 35 gene sets were significant at FDR < 25% and 99/165 gene sets were up-regulated in phenotype l, and 55 gene sets were significant at FDR < 25%. The top-five up-regulated gene sets were basal cell carcinoma (NES = 2.15, P = 0.000), dilated cardiomyopathy (NES = 2.11, P = 0.000), glycosaminoglycan biosynthesis chondroitin sulfate (NES = 2.08, P = 0.000), complement and coagulation cascades (NES = 2.06, P = 0.002), and hypertrophic cardiomyopathy hcm (NES = 1.91, P = 0.002), while the top-five downregulated gene sets were RNA degradation (NES = −2.08, P = 0.008), valine leucine and isoleucine degradation (NES = −2.06, P = 0.004), ubiquitin-mediated proteolysis (NES = −2.02, P = 0.006), propanoate metabolism (NES = −1.95, P = 0.006), and oocyte meiosis (NES = −1.92, P = 0.014) (Figs. 3C–3L). The detailed GSEA analysis is shown in Table S3.

Figure 3 Functional enrichment analysis of RBPs.

(A) TOP-10 terms of BP, CC and MF in up-regulated RBPs. (B) TOP-10 terms of BP, CC and MF in down-regulated RBPs. (C–L) TOP-5 GSEA analysis of the KEGG pathway in RBPs. RBPs, RNA-binding proteins. BP, Biological Processes, CC, Cellular Components, MF, Molecular Functions, KEGG, Kyoto Encyclopedia of Genes and Genomes, GSEA, Gene Set Enrichment Analysis.

PPI network construction and hub modules screening

We constructed a PPI network with 147 nodes and 543 edges based on the STRING database’s data using Cytoscape software to further explore the roles of differentially expressed RBP in CRC (Fig. 4A). The co-expression network was constructed by MODE Plug-in to the study potential hub modules and the top-four crucial modules. The compulsory modules 1, 2, 3 and 4 consisted of 18 nodes and 144 edges; nine nodes and 28 edges; five nodes and 10 edges; and six nodes and 12 edges, respectively (Fig. 4B). RBPs in the compulsory modules 1–4 were mainly enriched in ribosome biogenesis, ncRNA processing, mRNA processing, mRNA binding, defense response to a virus, response to a virus, DNA alkylation and DNA methylation. The PPI network is shown in Table S4.

Figure 4 Protein–protein interaction network and module analysis.

(A) Protein-protein interaction (PPI) network of differentially expressed RBPs. (B) Critical module from the PPI network. Blue circles: down-regulation RBPs (adjust P-value＜0.05), red circles: upregulation RBPs (adjust P-value＜0.05). RBPs, RNA-binding proteins.

Construction prognosis model of RBPs and survival analysis

One hundred forty-seven differently expressed RBPs were identified from the PPI network. To further exploring the prognostic value of these RBPs, we carried out a univariate Cox regression analysis and screened out 15 potential prognostic-associated key RBPs, including six Low-Risk BAPs (HR < 1, P < 0.05) and nine High-Risk RBPs (HR > 1, P < 0.05, Fig. 5A). Nine RBPs were filtered out by multivariate COX regression analysis (Fig. 5B). The prognosis model was constructed based on these nine RBPs. The full name and coefficients of these proteins are shown in Table 1.

Figure 5 Risk score analysis of nine-RBPs prognostic model in the TCGA cohort.

(A) The hazard ratios calculated by univariate Cox regression. (B) The hazard ratios calculated by multivariate Cox regression analysis. (C) Kaplan-Meier survival analysis of CRC patients for high and low risk subgroups. (D) ROC analysis of the specificity and sensitivity of the OS based on the risk score. (E) Heatmap of the BAPs expression profiles. (F, G) The distribution of patient’s survival time and survival status. The black dotted line is the optimum cutoff dividing patients into high risk and low risk groups. (H) Univariate independent prognosis Cox regression analysis. Forest plot of the relationship between risk factors and OS. (I) Multivariate independent prognosis Cox regression analysis. Forest plot of the association between risk factors and OS. ROC, Receiver Operating Characteristic, OS, Overall survival.

Table 1 Prognosis-related RBPs (multivariate Cox regression analysis) and prognosis model coefficient.

RBPs	Official full name	Coefficient	HR	Lower 95% CI	Upper 95% CI	P- value	
PNLDC1	PARN Like, ribonuclease domain containing 1	0.2943	1.342146139	1.093946483	1.646658486	0.004792852	
RRS1	Ribosome biogenesis regulator 1 homolog	0.4835	1.621755576	1.170686306	2.246623313	0.003641482	
HEXIM1	HEXIM P-TEFb complex subunit 1	0.3609	1.434554531	1.045118889	1.969102963	0.025545301	
PPARGC1A	PPARG coactivator 1 alpha	−0.1757	0.838828662	0.668974465	1.05180924	0.127906411	
PPARGC1B	PPARG coactivator 1 beta	−0.3660	0.693496854	0.464111999	1.036253939	0.074071497	
BRCA1	BRCA1 DNA repair associated	−0.4157	0.659848664	0.428493546	1.016118593	0.059110143	
CELF4	CUGBP Elav-like family member 4	0.5460	1.72633526	1.307387604	2.279533186	0.000118191	
AEN	Apoptosis enhancing nuclease	0.4375	1.548761254	1.169130348	2.051662954	0.002295159	
NOVA1	NOVA alternative splicing regulator 1	0.1639	1.178115785	0.957878215	1.448990884	0.120567185	

The risk score for each patient was calculated based on the following formula:

Risk score = 0.2943 × expression level of PNLDC1 + 0.4835 × expression level of RRS1 + 0.3609 × expression level of HEXIM1 + (−0.1757) × expression level of PPARGC1A + (−0.3660) × expression of PPARGC1B + (−0.4157) × expression level of BRCA1 + 0.5460 × expression level of CELF4 + 0.4375 × expression level of AEN + 0.1639 × expression level of NOVA1.

The risk score was applied to predict prognosis, and 586 CRC patients were divided into high-risk and low-risk subgroups according to the median risk score. Kaplan–Meier cumulative curves showed a significant difference in OS between the high-risk and low-risk groups. Patients with high-risk scores had a worse OS than those with low-risk scores (Fig. 5C). A ROC analysis was used further to assess the prognostic accuracy of the prognosis model. Our results indicated that the AUC value was 0.712 (Fig. 5D), which showed a moderate diagnostic ability. An elevated risk score was correlated with a decreased survival time and an increase in mortality (Fig. 5E). Age, clinical-stage, primary tumor size (T), regional lymph node involvement (N), distant metastasis (M) and risk score were significantly associated with OS in the univariate independent prognostic analysis (P < 0.001, Fig. 5F; Table 2). The multivariate independent predictive analysis demonstrated that age, T, and risk scores were also independent prognostic predictors (P < 0.05, Fig. 5G; Table 2). We assessed whether the nine-RBPs predicting model had an equivalent prognostic value in other CRC cohorts and used the same formula in the validation group in which data come from the GSE39582 and GSE87211 datasets. The result showed that patients with high-risk scores also had a worse OS than those with low-risk scores; the risk score was also an independent prognostic predictor in the GSE39582 and GSE87211 cohorts (Fig. 6; Table 3). These results showed that the prognostic model, based on the nine RBPs, had higher sensitivity and specificity.

Table 2 The prognostic value of different clinical characters in the train group.

	Univariate prognostic analysis	Multivariate prognostic analysis	
	HR	95% CI	P-value	HR	95% CI	P-value	
Age	1.0360	[1.0158–1.0567]	4.49E−04	1.0529	[1.0316–1.0747]	7.85E−07	
Gender	1.0782	[0.7154–1.6251]	0.7189	0.8547	[0.5624–1.2989]	0.4621	
Stage	2.4378	[1.9233–3.0901]	1.75E−13	1.7652	[0.8994–3.4643]	0.0986	
T	2.9977	[1.9876–4.5212]	1.64E−07	1.6070	[1.0023–2.5766]	0.0489	
M	4.8650	[3.1979–7.4012]	1.47E−13	1.5751	[0.6184–4.0119]	0.3408	
N	2.0793	[1.6380–2.6396]	1.81E−09	1.1524	[0.7623–1.7422]	7.54E−04	
Riskscore	1.0843	[1.0566–1.1127]	8.63E−10	1.0827	[1.0496–1.1168]	5.08E−07	

Figure 6 Risk score analysis of nine-RBPs prognostic model in the GEO cohort.

(A) Kaplan-Meier survival analysis of CRC patients for high and low risk subgroups. (B) ROC analysis of the specificity and sensitivity of the OS based on the risk score. (C) Heatmap of the RBPs expression profiles. (D and E) The distribution of patient’s survival time and survival status. The black dotted line is the optimum cutoff dividing patients into high risk and low risk groups. (F) Univariate independent prognosis Cox regression analysis. Forest plot of the relationship between risk factors and OS. (G) Multivariate independent prognosis Cox regression analysis. Forest plot of the association between risk factors and OS. ROC, Receiver Operating Characteristic, OS, Overall survival.

Table 3 The prognostic value of different clinical characters in the validation group.

	Univariate prognostic analysis	Multivariate prognostic analysis	
	HR	95% CI	P-value	HR	95% CI	P-value	
Age	1.0238	[1.0115–1.0362]	1.32E−04	1.0317	[1.0191–1.0445]	6.45E−07	
Gender	1.1140	[0.8394–1.4785]	0.4547	1.2753	[0.9587–1.6963]	0.0948	
Stage	1.9205	[1.5559–2.3704]	1.23E−09	0.7029	[0.4540–1.0884]	0.1140	
T	1.9591	[1.4880–2.5794]	1.65E−06	1.6453	[1.2436–2.1767]	0.0005	
M	5.3710	[3.7963–7.5989]	2.20E−21	6.9200	[3.6552–13.1009]	2.85E−09	
N	1.4507	[1.2115–1.7371]	5.20E−05	1.5135	[1.1301–2.0269]	0.0054	
Riskscore	1.2161	[1.0965–1.3487]	2.12E−04	1.1347	[1.0056–1.2805]	0.0403	

Construction of a nomogram based on the nine crucial RBPs

A nomogram can quantitatively identify an individual’s clinical risk by combining multiple risk factors (Liang et al., 2015; Won et al., 2015). We used the nomogram to predict the probability of 1-, 3- and 5-year OS by merging the nine RBPs signatures for TCGA-CRC and verified these results using the GEO data sets (Fig. 7A). The distribution point of each RBPs was proportional to its risk contribution to the survival and was normalized to the distribution of 0–100. We calculated each patient’s total points by summarizing the number of points for all RBPs and estimated the survival rates by drawing a vertical line between each prognosis axis and the total point axis. This tool may help practitioners make clinical decisions for CRC patients. The calibration curve demonstrated that the actual survival rate matched the predicted 1-, 3- and 5-year survival rate. The C-index was 0.77 in the TCGA database (Fig. 7B). The nomogram was verified in the GEO cohorts, the C-index was 0.73, and the 1-, 3- and 5-year calibration curves are shown in Fig. 7C.

Figure 7 The nomogram to predict 1-, 3- and 5-year OS of CRC patients.

(A) The nomogram for predicting 1-, 3- and 5-year OS of CRC by nine RBPs signature. The Calibration curves for predicting 1-, 3- and 5-year OS of TCGA-CRC (B, D and F) and GEO-CRC (C, E and G). OS, Overall survival.

Co-expression analysis of RBPs

To explore more potential prognosis proteins, we used the nine RBPs in the model for co-expression analysis. CELF4 and LUZP4 were co-expressed; NOVA1 was co-expressed with RBFOX3, ELAVL4, DZIP1, SAMD4A, ZCCHC24, AFF3, RBM20, ENOX1, RBPMS2 and CPEB1. PPARGC1A was co-expressed with RBM47, RAVER2 and CPEB3. PPARGC1B was co-expressed with RBM47, RAVER2 and LRRFIP2. RRS1 was co-expressed with BOP1 (∣correlation coefficient∣＞0.6, P＜0.001, Fig. 8). The co-expression analysis is shown in Table S5.

Figure 8 The Co-expression analysis of RBPs.

(A) The proteins in the left box are RBPs, and in the right box are co-expressed proteins with RBPs. (B) CELF4 was positively co-expressed with LUZP4. NOVA1 was positively co-expressed with AFF3 (C) CPEB1 (D) DZIP1 (E) ELAVL4 (F) ENOX1 (G) RBFOX3(H) RBPMS2 (I) RBM20 (J) SAMD4A (K) ZCCHC24 (L). PPARGC1A was positively co-expressed with CPEB3 (M) RAVER2 (N) RBM47 (O). PPARGC1B was positively co-expressed with LRRFIP2 (P) RAVER2 (Q) RBM47 (R). RRS1 was positively co-expressed with BOP1 (S).

Validation of the expression of hub RBPs

The IHC results of the HPA database were used to explore the expression of the nine hubs RBPs in CRC, and the result showed that RRS1 and BRCA1 were highly expressed in the CRC tissues compared with normal tissues. However, PNLDC1 was more highly expressed in the normal colorectal tissues than in the cancer tissues. HEXIM1 was highly expressed in both normal and colorectal tissues. In contrast, CELF4 was not expressed in normal and colorectal tissues. Furthermore, the protein expression of NOVA1 was not significantly different between normal and CRC tissues (Fig. 9).

Figure 9 Validation of hub RBPs expression in CRC and normal colorectal tissue in the HPA database.

(A) BRCA1, (B) CELF4, (C) HEXIM1, (D) NOVA1, (E) PNLDC1, (F) RRS1.

Discussion

RNA regulatory-mediated RBPs are involved in maintaining homeostasis and in cancer progression. However, the tumor-related functions of most RBPs and the detailed mechanisms of their anti-tumor effects remain to be explored (Jonas, Calin & Pichler, 2020; Konishi et al., 2020). We screened out 178 differently expressed RBPs in normal and tumor tissues from the TCGA–CRC data set. We then comprehensively analyzed the appropriate BP and pathways and constructed a PPI network and co-expression network of these RBPs. We applied univariate and multivariate COX regression analysis, survival analysis, and ROC analyses and constructed a nomogram to predict prognoses and further explore the 10 RBPs related to the clinicopathological characteristics of CRC patients. We constructed a risk model based on nine prognostic-associated core RBP genes to predict the prognosis of CRC. These discoveries may provide novel biomarkers to use in the diagnosis and prognosis of CRC patients.

The results of our functional pathway enrichment analysis showed that differently expressed RBPs were abundantly enriched in the ncRNA metabolic process, ncRNA processing, ribonucleoprotein complex biogenesis, mRNA processing, regulation of translation, nucleolar part, endolysosome membrane, apical dendrite, catalytic activity, RNA activity, tRNA binding, mRNA binding, AU-rich element binding and mRNA 3′-UTR AU-rich region binding. Previous studies demonstrated that RBPs regulate various aspects of RNA metabolism and tumorigenesis (Mitobe et al., 2020; Zhu, Rooney & Michlewski, 2020). The ncRNA regulates proliferation checkpoints and inflammatory gene expression in CRC (Ma, Long & Huang, 2019). The ribonucleoprotein complex can promote excessive mitosis and the growth of CRC cells (Konishi et al., 2020). mRNA processing, tRNA binding, and mRNA binding play an essential role in CRC growth and metastasis (Kong & Wang, 2020; Wang et al., 2019a). KEGG pathways analysis showed that the aberrantly expressed RBPs also regulated basal cell carcinoma, dilated cardiomyopathy, glycosaminoglycan biosynthesis, chondroitin sulfate, complement and coagulation cascades, and hypertrophic cardiomyopathy.

We constructed a PPI network based on differently expressed RBPs and obtained four key modules, including 38 core RBPs. Many of these essential RBPs have been shown to play a crucial role in the development and progression of cancers. BOP1 is a direct transcription target of β-catenin/TCF4, which induced cell migration and experimental metastasis of CRC cells (Qi et al., 2016). DKC1 may regulate CRC angiogenesis and metastasis by directly activating HIF-1α transcription (Hou et al., 2020). Ji et al. (2019) showed that PTBP2 could elevate translational levels of RUNX2, which plays a vital role in CRC metastasis. NOVA1 expression controls the proliferation and invasive properties of CRC cells by enhancing IL-6/JAK2/STAT3 signal transduction, and in turn, up-regulating matrix metalloproteinases (MMPs) 2, 7 and 9 (Hong et al., 2019). CRC associated with inflammation by activation initiated TLR3 and TLR7 (He et al., 2017). EZH2 expression and activity are associated with colorectal carcinogenesis and are most often expressed in highly intraepithelial lesions (Bremer et al., 2021). The PPI network’s module analysis showed that CRC is related to ribosome biogenesis, ncRNA processing, ribonucleoprotein complex biogenesis, ncRNA metabolic process, mRNA processing, mRNA binding and RNA splicing.

Nine RBPs, including PNLDC1, RRS1, HEXIM1, PPARGC1A, PPARGC1B, BRCA1, CELF4, AEN and NOVA1, were identified as prognosis-associated core RBPs based on multivariate Cox regression analysis. Previous studies reported that the expression of RRS1 may promote the development of colon cancer (Wu et al., 2017) and that HEXIM1 is a positive regulator of p53 (Lew et al., 2012). PPARGC1A and PPARGC1B have also been demonstrated to contribute to CRC susceptibility (Lin et al., 2019). Studies have also shown that BRCA1 mutation carriers are not at high risk for CRC (Cullinane et al., 2020) and that NOVA1 is expressed at higher levels in CRC cell lines (Hong et al., 2019). These RBPs are related to tumorigenesis and progression in CRC patients, which are in-line with our results.

We constructed a risk model to predict CRC prognosis through multivariate Cox analysis based on the nine hub RBPs coding genes. The TCGA cohort was used as the training group. ROC curve analysis showed that the nine gene markers had a better diagnostic ability and could select CRC patients with poor prognosis and AUC values of ROC, which was consistent with previous reports (Jeun et al., 2019). Moreover, the validation group results (GEO cohorts) were consistent with the results from the training group. However, the molecular mechanisms of the nine RBPs contributing to CRC are still poorly understood, and further investigation into their potential mechanisms is needed.

We developed a nomogram to more accurately predict 1-, 3- and 5-year OS. In addition to traditional clinicopathological characteristics, including tumor size, TNM stage, and tissue subtypes, gene expression can also be incorporated into predictive nomogram models to predict clinical outcomes (Reichling et al., 2019; Sjoquist et al., 2018). A nomogram for predicting 3- and 5-year recurrence-free survival in NSCLC has been reported and included a prognostic score calculated from the autophagy gene signature (Liu et al., 2019). A combination of autophagy gene markers with a prognostic element has a better prognostic value than an individual marker (Mo et al., 2019). The calibration curve showed that a nomogram could accurately predict 3- and 5-year survival probabilities (Xiong et al., 2017). We were able to show that a nomogram, including the nine RBPs signature, could better predict 1-, 3- and 5-year survival possibility of CRC patients. These results suggest that the nine-RBPs prognostic model is of value in determining the treatment plan of CRC patients.

Co-expression analysis indicated that five RBPs were co-expressed with 16 proteins. BOP1 is related to cell migration and experimental metastasis of CRC cells (Qi et al., 2016). The high expression of CPEB3 is related to the poor prognosis of CRC (Waku et al., 2020). ELAVL4 is up-regulated in CRC patients (Huang et al., 2012). LRRFIP2 is associated with hereditary non-polyposis CRC (Lynch Syndrome) (Morak et al., 2011). The down-regulation of RBM47 during CRC progression may promote EMT and metastasis (Rokavec et al., 2017). The effects of other proteins, including AFF3, CPEB1, DZIP1, ENOX1, LUZP4, RAVER2, RBFOX3, RBM20, RBPMS2, SAMD4A and ZCCHC24, in the development of CRC is poorly understood and should be investigated further.

Our RBPs prognostic model reduces the cost of testing and is able to be applied clinically. The nine RBPs-prediction model has better survival prediction capability in CRC patients. Nevertheless, this study has several limitations. Firstly, we designed the study based on a retrospective analysis, and prospective studies are needed to validate the future results. Secondly, the data set does not provide some clinical information, such as tumor grade in CRC, which may reduce the statistical effectiveness and dependability of multivariate Cox analysis.

Conclusions

We systematically investigated the prognostic value of RBPs in CRC using multiple bioinformatics analyses. These RBP may be related to the occurrence, development, invasion and metastasis of CRC. The prognostic model of nine hubs RBPs was established, and the risk score was shown to be an independent prognostic factor for CRC. Our results will help to understand the pathogenesis of CRC and develop new therapeutic targets and prognostic biomarkers. These findings provide a comprehensive perspective for further study of the role of the hub RBPs in the pathogenesis of CRC and identify potential molecular markers for the diagnosis and treatment of CRC.

Supplemental Information

Supplemental Information 1 Characteristics of TCGA colorectal cancer cohort and GEO dataset.

Click here for additional data file.

Supplemental Information 2 The detailed information of GO terms.

Click here for additional data file.

Supplemental Information 3 The detailed information of GSEA analysis.

Click here for additional data file.

Supplemental Information 4 The detailed information of the PPI network.

Click here for additional data file.

Supplemental Information 5 The detailed information of the co-expression analysis.

Click here for additional data file.

Supplemental Information 6 Raw data and code.

Click here for additional data file.

We are grateful for the availability of the data from the TCGA and GEO databases.

The data of clinical for this study can be found in the TCGA database. The database of validation group for this study can be found in the Gene Expression Omnibus (GEO) database, including GSE39582 and GSE87211 datasets. The data were obtained from TCGA and GEO database, strictly following the publication guidelines of TCGA and GEO. Other data used and/or analyzed during the current study are available from the first or corresponding author on reasonable request.

Additional Information and Declarations

Competing Interests

Author Contributions

Data Availability

The authors declare that they have no competing interests.

Yandong Miao conceived and designed the experiments, performed the experiments, analyzed the data, prepared figures and/or tables, authored or reviewed drafts of the paper, and approved the final draft.

Hongling Zhang conceived and designed the experiments, performed the experiments, analyzed the data, prepared figures and/or tables, authored or reviewed drafts of the paper, and approved the final draft.

Bin Su conceived and designed the experiments, performed the experiments, analyzed the data, prepared figures and/or tables, authored or reviewed drafts of the paper, and approved the final draft.

Jiangtao Wang performed the experiments, analyzed the data, prepared figures and/or tables, and approved the final draft.

Wuxia Quan analyzed the data, prepared figures and/or tables, and approved the final draft.

Qiutian Li analyzed the data, prepared figures and/or tables, and approved the final draft.

Denghai Mi conceived and designed the experiments, prepared figures and/or tables, authored or reviewed drafts of the paper, and approved the final draft.

The following information was supplied regarding data availability:

The raw measurements are available in the Supplemental Files.

The clinical data for this study are available in the TCGA database (https://portal.gdc.cancer.gov/).

The data for the validation group is available in the Gene Expression Omnibus (GEO, https://www.ncbi.nlm.nih.gov/geo/), including GSE39582 and GSE87211 datasets.

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
