# Peer review of "Construction and validation of an RNA-binding protein-associated prognostic model for colorectal cancer"

_PeerJ, doi:10.7717/peerj.11219_

## Round 0.1 · original submission · Major Revisions

It seems the paper replicates most of the methodology, results, figures from the following two papers:

1) Integrated analysis of RNA-binding proteins in human colorectal cancer | World Journal of Surgical Oncology | Full Text (biomedcentral.com)

2) Frontiers | Molecular Characterization and Clinical Relevance of RNA Binding Proteins in Colorectal Cancer | Genetics (frontiersin.org)

Before starting the revision, consider that you have to cite these papers AND explain how your research was conducted differently.

PeerJ asks for original research papers!

Reviewer 1 ·

Basic reporting

No comment

Experimental design

Line 109: Please briefly describe the inclusion and exclusion criteria
Please add the IHC method under the Method Section
What’s your clinical trial registry?

Validity of the findings

Line 374: The database provided the association in a cross-sectional manner, which does not show causality. So the RBPs are just showing diagnostic/prognostic functions. Further studies are warranted to determine their therapeutic effects.

The authors have identified multiple RBPs that are correlated with CRC progression. Is there any RBP or couple of RBPs you want to highlight?

Additional comments

The authors provided novel and extensive evidence showing that several RBPs are differentially expressed in CRC vs non-CRC tissues, and could be potential biomarkers of CRC progression. The study was well-designed and correctly translated.

·

Basic reporting

This paper is well-organized with background material that can be improved. There can be several improvements that can be made to the manuscript to improve basic reporting and overall usefulness.

a) minor Grammar Issues
1) "shown that that" should be "shown that " on line 89
2) "are still poorly understand" on line 339 should be "are still poorly understood"
3) missing articles at some places throughout the paper.

b) Background context

1) "the role of RBPs in tumor development is relatively unknown" is pretty bold since there are multiple studies published over last few years

b) I think authors have skipped some background information about various studies done recently which studied the role of RBC's in CRC
Integrated analysis of RNA-binding proteins in human colorectal cancer | World Journal of Surgical Oncology | Full Text (biomedcentral.com)


c) Figures and tables

1) Authors should improve legibility of text and resolution in all figures except figure-4

2) Risk Score should be mathematically written

Experimental design

Experimental design
The article Fits well within the Aims and Scope of the journal.

1) Did the Authors perform some variable selection or regularization to reduce overfitting in the case of multivariate cox regression? Did they select a model just based on AIC criteria

2) Is there any rationale for selecting mentioned GSE studies as a similar methodology is used in different studies :
Integrated analysis of RNA-binding proteins in human colorectal cancer | World Journal of Surgical Oncology | Full Text (biomedcentral.com)
Frontiers | Molecular Characterization and Clinical Relevance of RNA Binding Proteins in Colorectal Cancer | Genetics (frontiersin.org)

Validity of the findings

No comments

---

## Round 0.2 · accepted · Accept

The authors properly performed the revision of the manuscript.

Reviewer 1 ·

Basic reporting

No comment

Experimental design

No comment

Validity of the findings

No comment

Additional comments

The revised manuscript has been improved significantly.

·

Basic reporting

The authors have addressed my comments

Experimental design

The authors have addressed my comments

Validity of the findings

The authors have addressed my comments

Additional comments

Thanks for citing those two papers. Can you please include how your method differs from the below two papers in your introduction section( not just the rebuttal)

1) Integrated analysis of RNA-binding proteins in human colorectal cancer | World Journal of Surgical Oncology | Full Text (biomedcentral.com)

2) Frontiers | Molecular Characterization and Clinical Relevance of RNA Binding Proteins in Colorectal Cancer | Genetics (frontiersin.org)